# The Scientific Information Model of Chang’e-4 Visible and Near-IR Imaging Spectrometer (VNIS) and In-Flight Verification

**DOI:** 10.3390/s19122806

**Published:** 2019-06-22

**Authors:** Chunlai Li, Zhendong Wang, Rui Xu, Gang Lv, Liyin Yuan, Zhiping He, Jianyu Wang

**Affiliations:** Key Laboratory of Space Active Opto-Electronics Technology, Shanghai Institute of Technical Physics, Chinese Academy of Sciences, Shanghai 200083, China; lichunlai@mail.sitp.ac.cn (C.L.); wangzhendong@mail.sitp.ac.cn (Z.W.); xurui@mail.sitp.ac.cn (R.X.); lvgang@mail.sitp.ac.cn (G.L.); yuanliyin@mail.sitp.ac.cn (L.Y.)

**Keywords:** the Chang’e-4 lunar rover, phase-locked, signal-to-noise ratio, spectral resolution, infrared focal plane components

## Abstract

The Chang’e-4 (CE-4) lunar rover, equipped with a visible and near-IR imaging spectrometer (VNIS) based on acousto-optic tunable filter spectroscopy, was launched to the far side of the moon on December 8, 2018. The detection band of the VNIS ranges from 0.45 to 2.4 μm. Because of the weak reflection of infrared radiation from the lunar surface, a static electronic phase-locked acquisition method is adopted in the infrared channel for signal amplification. In this paper, full-link simulations and modeling are conducted on the infrared channel information flow of the instrument. The signal characteristics of the VNIS are analyzed in depth, and the signal to noise ratio (SNR) prediction and laboratory verification are presented. On 4 January 2019, the VNIS started working successfully and acquired high-resolution spectrum data of the far side of the moon for the first time. Through analysis we have found that the SNR ratio is in line with our predictions, and the data obtained by VNIS in orbit are consistent with the information model proposed in this paper.

## 1. Introduction

The Chang’e-4 (CE-4) lunar rover is the first man-made spacecraft launched to the far side of the moon, and its rover (Yutu-2) is equipped with a visible and near-IR imaging spectrometer (VNIS), which is used to analyze the composition of lunar surface minerals [1,2]. The VNIS inherits information from the Chang’E-3 Lunar Rover’s scientific payloads, and the software is optimized. The VNIS is a spectrum detector based on acousto-optic tunable filter (AOTF) spectroscopy [3,4], and it has two detection channels—visible near-infrared (450–950 nm) and short-wave infrared (900–2400 nm). It performs spectral analysis and imaging detection of minerals on the lunar surface under appropriate solar illumination and assists in the comprehensive detection of minerals and chemical compositions in the patrol areas. 

The AOTF spectrometer has been used in deep space exploration for a long time. The SPICAM [5], which was loaded on the ESA Mars Express mission (2003), was one of the first spectroscopic applications of the AOTF on a civilian spacecraft. It is a single-pixel spectrometer which can sequentially measure the spectrum of reflected solar radiation from Mars between 1.1 and 1.7 µm with spectral resolution of 3.5 cm^−1^, and spatial resolution of ~6 km from 250 km of the Mars express orbit. Similar instruments, such as the SPICAV [6], were loaded on the ESA Venus express mission (2005). NASA has also been working on spectrometers based on the AOTF. AIMS [7] is a compact, two-channel AOTF imaging spectrometer developed by NASA/GSFC, and it is one of the alternative payloads of the NASA Mars rover. Since 2006, the Shanghai Institute of Technical Physics began to study imaging spectrometers based on AOTF. The VNIS is a payload of the lunar rover for 0.45–2.4 μm spectral bands detection and will be able to inspect and probe minerals for the rover region [2] (China’s Chang’e 3 and Chang’e 4 unmanned lunar rover).

The CE-4 spacecraft was successfully launched on 8 December 2018, and it landed on the moon on 3 January 2019. Then the VNIS was powered on and acquired infrared spectral data of the lunar surface. This paper focuses mainly on simulation analyses and laboratory tests based on the infrared channel design features and information flow model of the instrument and carries out verification according to the acquired in-orbit data.

## 2. Instrument Description

### 2.1. Basic Principle of The Visible and Near-IR Imaging Spectrometer (VNIS)

The VNIS uses an acousto-optic tunable filter (AOTF) for the spectral filtering of light. When a beam of broadband light passes through an optically elastic crystal that vibrates at high frequencies, monochromatic light of a certain wavelength will be diffracted inside the crystal and transmitted from it at an angle, and the non-diffracted light travels through the crystal directly in the original direction, thereby achieving the goal of light filtering [8,9]. 

The AOTF is a spectroscopic device based on the principle of acousto-optic diffraction and is composed of a crystal and a transducer bonded on it [10,11]. The transducer converts the electric signal into ultrasonic vibrations in the crystal, which generate spatially periodic modulations. When incoming light is incident on the AOTF, the wavelength of the diffracted light is related to the frequency of the driving electric signal, and it can be changed by varying the frequency of the electric signal [12]. 

According to this principle, by changing the driving frequency of the AOTF crystal through rapid scanning, the wavelength of the first-order diffraction light passing through the AOTF changes sequentially. In this way, the spectral information of the target can be obtained by VNIS [10]. The VNIS can obtain a spectral image in the VIS/NIR channel (visible and near-infrared waveband) and spectral data in the SWIR channel (short waveband) simultaneously. Figure 1 shows the components and basic principle of the VNIS, and Table 1 presents the main specification of the VNIS. 

The VNIS has two AOTFs, one for the VIS/NIR channel, and the other for the SWIR channel. Each AOTF is about the size of a matchbox. The two AOTFs were developed by the 26th Research Institute of China Electronic Science and Technology Group Corporation. Each AOTF has two RF (Radio Frequency) connectors, one for high frequency and the other for low frequency. Table 1 also shows the relationship between the driving frequencies for AOTFs and the corresponding diffraction wavelength.

In order to improve the sensitivity of the SWIR channel of the VNIS, a static electronic phase-locked acquisition method is adopted to realize the high sensitivity in the SWIR channel.

The components and basic principle of the VNIS are shown in Figure 1. The VNIS is equipped with a calibration unit at the light entrance. The calibration unit consists of an ultrasonic motor, framework, and diffuser panel inside. The calibration unit could be located at a different position by the ultrasonic motor for lunar surface detection, in-orbit calibration and dust-proof functions. While the Chang’4 spacecraft was operating in-orbit flight and the soft-landing phase, the calibration unit was locked to thermal insulation and it was dust-proof. After the Yutu-2 rover separated from the lander on the moon, the VNIS was remote controlled to collect soil images and spectrum and the calibration unit was set to be fully open in order not to block the reflectance light into the imaging lens. The angle between the framework of calibration unit and horizontal mounting face of the VNIS was about 55°. When the VNIS was required to perform an in-orbit calibration, the calibration unit would be located at a horizontal position by the hall sensor and the sun light could be diffused into the VNIS from diffuser panel inside. The structure and function of the calibration unit is shown in Figure 2.

### 2.2. Optical Design of the SWIR Channel

The optical system of the VNIS in the infrared channel is shown in Figure 3. The target input rays enter into the instrument through an imaging lens, after being collimated into a parallel beam, which drives the AOTF to work, so that the emitted light passing through the AOTF forms an infrared monochromatic light of a specific wavelength and then converges to the detector through a convergent lens. 

According to the basic principle of AOTF spectrophotometry, after passing through the AOTF, three light rays are formed through the convergent lens, which are positive first-order diffraction light, negative first-order diffraction light and zero-order diffraction light [10,11]. An infared(the material is InGaAs) detector with a diameter of 1 mm is placed at the convergence of the positive first-order diffraction light, and a light filter is designed for the zero-order diffraction light, thereby suppressing the stray light.

### 2.3. The Information Link of Infrared Channel

Figure 4 shows the block diagram of the SWIR channel of the VNIS, which is the basis for building the infrared information flow model. In order to make the AOTF crystal work in combination with expectations, an radio frequency (RF) signal of a specific frequency (between 42.6 MHz to 117.7 MHz) should be applied to it, which is generated by DDS(digital display scope) chip configured with FPGA(Field Programmable Gate Array) and then amplified by the RF power amplifier. After the AOTF crystal is driven by the RF signal, polychromatic light enters the AOTF and penetrates to produce three channels. In the actual design, we chose to detect the positive first-order diffraction light. Due to the low albedo of the lunar surface, the positive first-order diffraction light after AOTF is very weak. In order to improve the ability to detect such a weak infrared signal, a static electronic phase-locked acquisition method is adopted. 

As shown in Figure 4, by controlling the output RF signal amplified by the power amplifier, the periodic modulation of the output two channels of monochromatic infrared light at the AOTF outlet can be controlled. The modulation frequency is 500 Hz. The positive first-order diffraction light after modulation is received by an InGaAs infrared detector, and the corresponding voltage signal is obtained through current-voltage conversion. The signal is processed by a phase-locked amplifier and after low-pass filtering, sent to the ADC (analog-digital conversion) chip. It is processed by the FPGA control circuit, which performs on-chip multiple accumulative averaging and then uploads it to the load processor. Figure 5 shows a picture of the SWIR channel processing circuit and the InGaAs infrared detector.

## 3. Signal Flow Model Simulation and Testing

### 3.1. The Signal Acquisition Model of Infrared Spectral

The phase-locked amplification method of infrared channel electronics is introduced in Section 2.3. In this section, we further discuss the infrared channel information flow model and signal characteristics. The infrared spectral signal acquisition model is established based on the AOTF spectroscopic system, as shown in Figure 6.

For weak infrared signal processing technology with phase-locked amplification, moving parts (such as the modulating reticle and chopper) are generally used for modulation between the signal light and background, such as the spatial target infrared spectroscopy system [6]. In the VNIS, a phase-locked amplification model without moving parts is designed according to the characteristics of AOTF, as shown in Figure 6. By controlling the output RF signal amplified by the power amplifier, the periodic modulation of the output positive first-order diffraction light at the AOTF outlet can be controlled. The modulation frequency is designed to be 500 Hz. This 500 Hz frequency is generated by the FPGA chip, not by the moving parts (such as motors or galvanometers), it has a great advantage in reliability for space applications. 

The positive first-order diffraction light after passing through the AOTF crystal is received by the InGaAs detector. After current-voltage conversion by a preamplifier, it is sent to the subsequent stage for signal conditioning. As shown in Figure 6, for two consecutive wavelength signals (band λ_i_ and band λ_i+1_), after 500 Hz electronic modulation, what emerges is a periodic (500 Hz) infrared analog signal similar to a square wave after passing through the preamplifier. After passive band-pass and pre-amplification, the square wave appears to be an approximate sinusoidal signal, and the peak of the waveform reflects the intensity of the infrared monochromatic light of the wavelength. Then, it converts the signal into two signals of the same frequency and the same amplitude but with an heir phase differing by 180 degrees. 

Specifically, one chain is designed as a voltage follower circuit and the other is designed as a reverse circuit, thus realizing a phase shift of 180 degrees. The phase-locked circuit in Figure 5 realizes the chip selection of two signals. It uses a high-speed analog switch (ADG409) to realize the phase-locked output so that the signal becomes half the sinusoid shown in the figure. After low pass filtering (4.8 Hz), the signal is converted into a corresponding DC level signal, which is fed for AD sampling after voltage bias and post-stage amplification. 

In order to further improve the detection sensitivity, the signal of the same wavelength is collected eight times by ADC chip (AD976), and then the sampled digital number (DN) values from the eight times are averaged and conducted within the FPGA. For infrared full-spectrum detection of lunar surface targets, the time-division detection of different wavelengths is achieved by varying the frequency of the RF signal applied to the AOTF crystal. The VNIS has 300 infrared sampling bands. The acquisition time of a single band is 0.4 s, so the acquisition time of the full spectrum channel is approximately 2 min.

The above model describes the mechanism of the infrared channel signal generation of the VNIS. At the forefront of the model, the infrared detector produces a photo-generated signal current I(λ) due to receiving infrared light energy P(λ). The model aims to enhance the sensitivity of the infrared channel. Based on the above information flow model, the source of the photo-generated current I(λ) at the input end can be further analyzed so as to obtain the signal-to-noise ratio of the system.

For the lunar surface spectral detection model, the surface of the moon can be approximated as a Lambert body. The target energy *P*(Δ*λ*) received by the VNIS can be expressed as [13,14]:(1)$$P\left(\Delta \lambda \right)=\frac{1}{4}\cdot E\left(\lambda \right)\cdot \frac{A}{(F\#{)}^{2}}\cdot \tau_{o}\left(\lambda \right)\cdot \rho \left(\lambda \right)\cdot \sin\theta \cdot \Delta \lambda$$
where *E*(*λ*) represents the spectral irradiance of the sun near the lunar surface. A is the pixel area of detector, F# is the optical aperture, *τ*_0_(*λ*) is the optical system efficiency (include the efficiency of AOTF), *θ* is the solar elevation angle, Δ*λ* is the spectral resolution, and *ρ*(*λ*) is the reflectance factor. Here is also a concept that the actual energy received by the instrument is the spectral radiance B(*λ*) forced by the target. The relationship between the radiance B(*λ*) and irradiance *E*(*λ*) is [1,2]:(2)$$B\left(\lambda \right)=\frac{E\left(\lambda \right)\cdot \rho \left(\lambda \right)\cdot \sin\theta }{\pi }$$

Generally, the target energy P(*λ*) is determined after the instrument system parameters are determined. That is, given the target radiant power received by the InGaAs detector in the information flow model in Figure 6, the signal current I(*λ*) of the infrared detector response can be expressed as:(3)$$I\left(\Delta \lambda \right)=P\left(\Delta \lambda \right)\times R_{\lambda }$$
where R_*λ*_ represents the current response of detector. The VNIS uses the J23TE2-66C-R01M-2.6 infrared detector manufactured by Judson, and the peak current response rate is about 1.2 A/W. After the signal current of the detector response is determined, the noise current I_nλ_ can be calculated by the following equation.
(4)$$I_{n\lambda }=\sqrt{2\times q\times (I_{s\lambda }+I_{dark}+I_{black})\Delta f}$$
where *q* represents the electron charge, *I_dark_* denotes the dark current of the detector, *I_black_* denotes the current caused by the thermal background radiation, and Δ*f* is the bandwidth of the information processing circuit. Due to the rapid development of the detector technology, the dark current of the InGaAs infrared detector at a low temperature of around −40 °C is basically zero. The SNR can be simplified as:(5)$$\mathrm{SNR}\left(\lambda \right)=\frac{I_{s\lambda }}{I_{n\lambda }}=\frac{I_{s\lambda }}{\sqrt{2\times q\times I_{s\lambda }\Delta f}}=\sqrt{\frac{I_{s\lambda }}{2\times q}}\frac{1}{\sqrt{\Delta f}}$$

As can be seen, given the system design parameters and the selected infrared detector, the system SNR is directly related to the information processing bandwidth Δ*f* of the circuit. The system adopts the information flow model shown in Figure 6. Here, Δ*f* is actually the low-pass filtering 4.8 Hz mentioned in the Figure 6, and the system noise beyond ((500 − 2.4) Hz–(500 + 2.4) Hz) is filtered out. 

In contrast, if the direct signal acquisition method (without phase lock-in method) is adopted, the bandwidth Δ*f* is generally around 2000 Hz, which is the mechanism of increasing the SNR ratio. The Figure 7 shows the estimated SNR curve based on the static electronic phase-locked acquisition method (bandwidth is about 4.8 Hz), which is compared with the direct signal acquisition method (bandwidth is about 2000 Hz). 

As shown in Figure 7, there exists a step at 1380 nm. The reason is that the SWIR AOTF crystal has two driving frequency range (F-H and F-L), and the 1380 nm is exactly the frequency switching point of the SWIR AOTF crystal. It can be seen from the Figure 7 that the SNR is about 600 at 1.7 μm. 

The parameters used in the calculations are:
➢E = 217 W/m^2^/μm@1.7 μm; ➢A = 1 mm^2^; ➢F#: 2.8; ➢τ_0_ = 0.35@1.7 μm; ➢Δλ = 8 nm@1.7 μm; ➢R_λ_ = 1.2 A/W@1.7 μm; ➢θ= 15°; ➢ρ = 0.09@1.7 μm;➢B = 1.61 W/m^2^/μm/Sr@1.7 μm.

### 3.2. Laboratory Testing and Evaluation

After the information flow model is established, the signal characteristic and the SNR of VNIS are tested in the laboratory. Figure 8 is a photo of the laboratory testing, where the light source is a halogen lamp and the target is a diffuse reflection plate. Adjusting the intensity of the Halogen lamp can simulate the change of the spectral radiance forced by the diffuse reflection plate, and synchronous measurement is carried out by a commercial spectrometer (FieldSpec 4). 

When the spectral radiance at 1.7 μm measured by the FieldSpec 4 is about 1.61 W/m^2^/μm/Sr, the intensity of the halogen lamp is stabilized, and the test is started. Figure 9 shows the original digital number (DN) values of 300 infrared spectral bands, and the SNR is about 500@1.7 μm (the signal DN is 1600, and the noise DN is 3.2). This is slightly different from the value (SNR≈600@1.7 μm) given in Section 3.1, because the light source here is not the sunlight with an equivalent elevation angle of 15 degrees and some transmission errors caused by the commercial spectrometer.

In order to further validate the information flow model, the analog signals (at 1.7 μm) were captured by an oscilloscope. Figure 10 shows the signal waveforms of A, B, C and D in the model in Figure 6, and the characteristics of those waveforms are consistent with the predictions of the established information flow model. This result is basically consistent with the information flow model described in Figure 6.

## 4. In-Flight Test

On 4 January 2019, the instrument obtained the infrared spectrum data of the first scene on the far side of the moon. Figure 11 shows the scene of the first infrared spectrum on the far side of the moon. Through absolute lab and in-flight radiometric calibration, we can get the relationship between DN values and spectral radiance at the entrance pupil, which is consistent with B(λ) in Figure 11. The spectral radiance B(λ) divided by the solar irradiance E(λ), multiplied by π, and further divided by the cosine of the incident angle θ can obtain the reflectance factor ρ(λ) [1]. The DN values of the original infrared spectrum data of A Point and its full-spectrum SNR are show in Figure 12 and Figure 13, and Figure 14 gives the infrared spectral reflectance curve (900–2400 nm) at A point. In this test, the solar elevation angle is about 15 degrees. 

According to the obtained raw data, the signal-to-noise ratio at 1.7 μm is calculated to be around 470, which is basically consistent with the prediction in Section 3.1 (Figure 9). This test validates the proposed AOTF system infrared spectral information processing model based on phase-locked amplification technology and also verifies the high sensitivity of the lunar surface weak infrared signal detector. 

## 5. Conclusions

In view of the extremely weak infrared reflectance spectral signals on the lunar surface, we proposed an information flow model, on the basis of which the signal-to-noise ratio has been predicted. As of the end of February 2019, the VNIS has operated six times on the lunar surface and has acquired 12 sets of infrared spectral data. It is found that an average SNR of 300 can still be obtained when the lunar albedo is around 9% and the solar elevation angle is 15 degrees, which further verifies the effectiveness of the proposed method for weak infrared signal. The SNR values of in-flight testing are basically consistent with the predictions given in this paper. In China’s future deep space exploration programs, the method proposed in this paper will be helpful to further study the acquisition of weak infrared spectral information on the surface of planets.

## Figures and Tables

**Figure 1 sensors-19-02806-f001:**
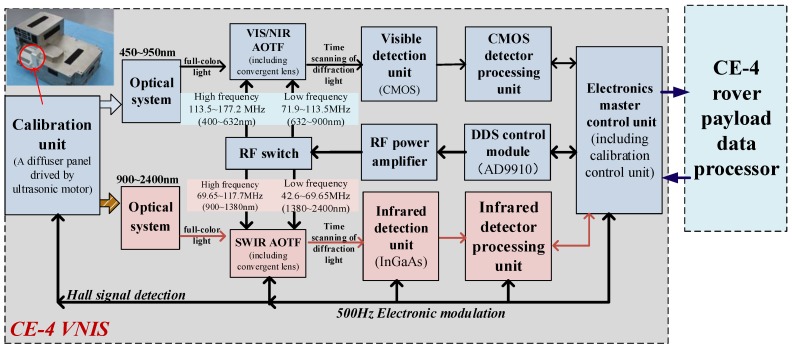
The components and basic principle of the VNIS.

**Figure 2 sensors-19-02806-f002:**
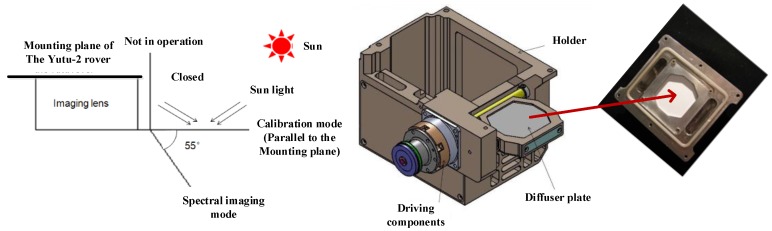
Structural and functional schematic view of the calibration unit.

**Figure 3 sensors-19-02806-f003:**
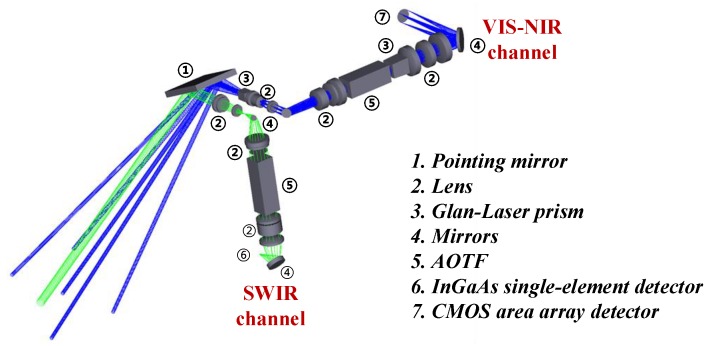
The optical design diagram of the VNIS.

**Figure 4 sensors-19-02806-f004:**
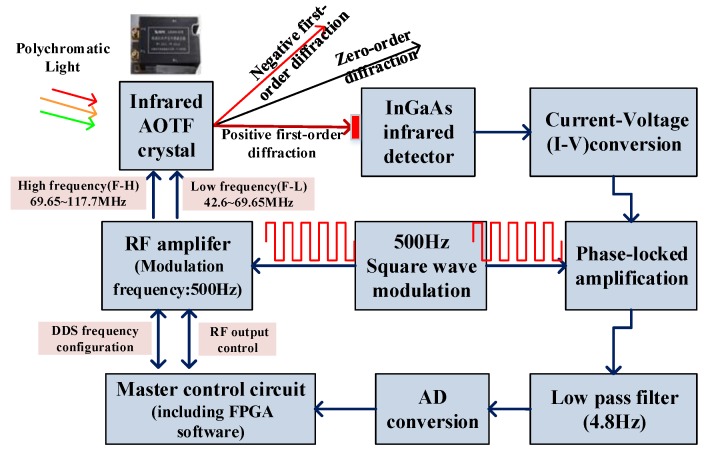
Schematic diagram of the phase-locked processing circuit of infrared channel.

**Figure 5 sensors-19-02806-f005:**
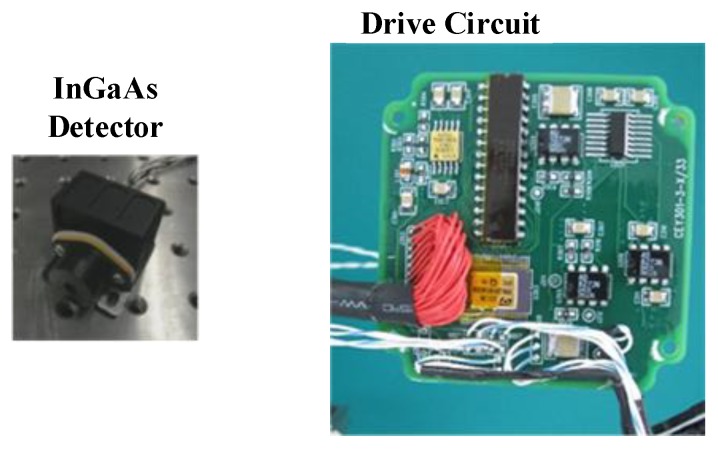
The picture of the SWIR channel processing circuit and the detector. The infrared detector type is J23TE2-66C-R01M-2.6, and is manufactured by Judson, with a peak current response rate of 1.2 A/W.

**Figure 6 sensors-19-02806-f006:**
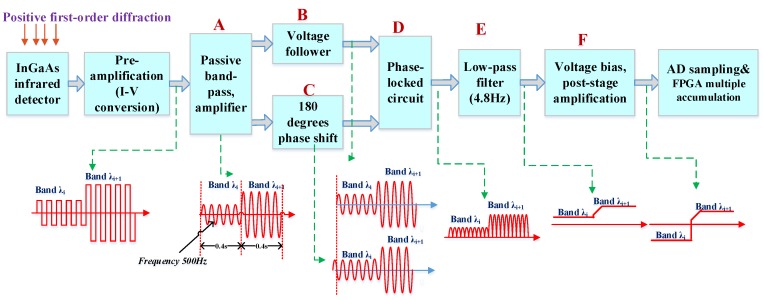
The information flow model of infrared spectrum acquisition based on acousto-optic tunable filter (AOTF) modulation.

**Figure 7 sensors-19-02806-f007:**
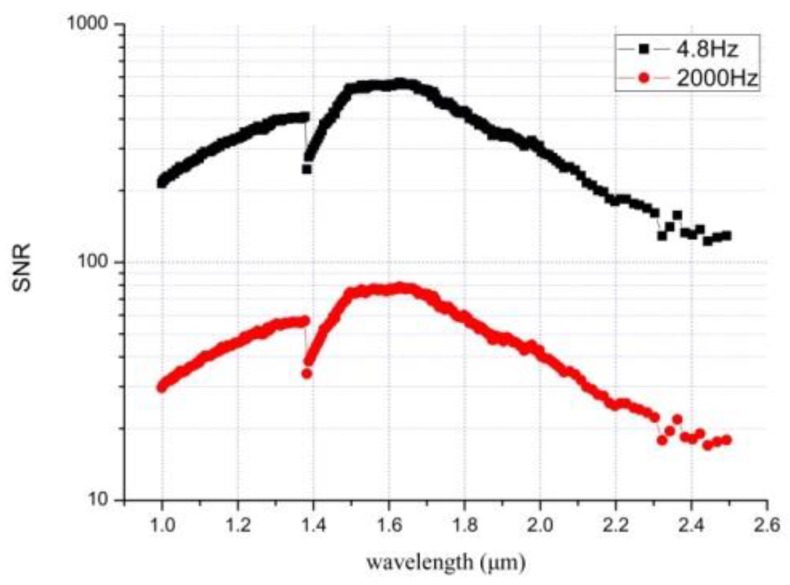
The signal-to-noise ratio (SNR) curve comparison between the static electronic phase-locked acquisition method (black) and the direct signal acquisition method (red).

**Figure 8 sensors-19-02806-f008:**
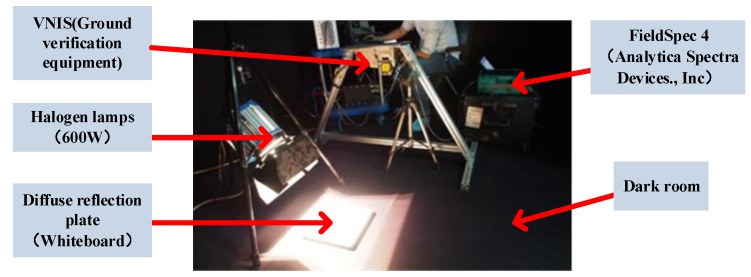
The experimental test was carried out in a dark room, using a halogen lamps to illuminate the diffuse reflection plate (reflectance >90%), and then at the same angle, using both VNIS (ground verification equipment) and a commercial spectrometer ((FieldSpec 4, Analytica Spectra Devices), Inc.) measure the spectral radiance forced by the diffuse reflection plate [15].

**Figure 9 sensors-19-02806-f009:**
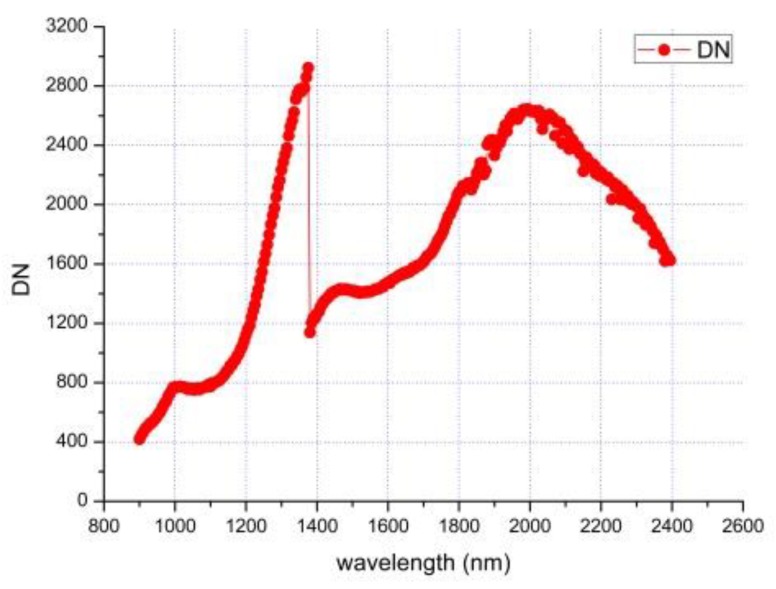
The original infrared full spectrum signal measured in the laboratory.

**Figure 10 sensors-19-02806-f010:**
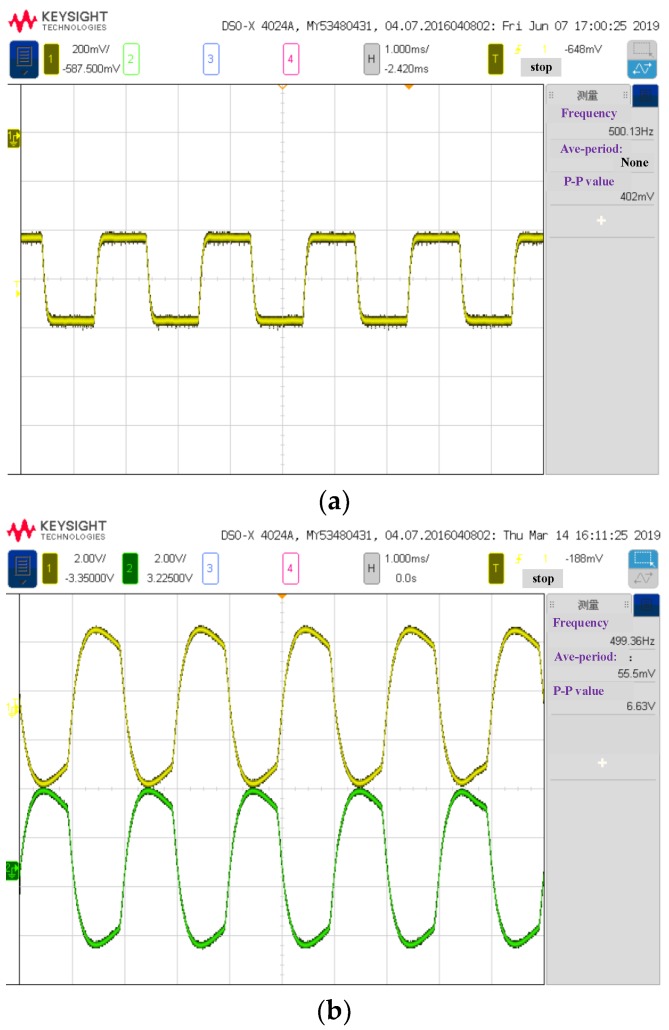
The analog signals of VNIS at 1.7 μm captured by oscilloscope (corresponding to A, B, C and D in Figure 6). (**a**) The analog signal after current to voltage(I-V) conversion captured by oscilloscope (point A in Figure 6); (**b**) The analog signal before phase-locked circuit captured by oscilloscope (the yellow is for point B in Figure 6, and the green is for point C in Figure 6); (**c**) The analog signal after the phase-locked circuit captured by oscilloscope (point D in Figure 6).

**Figure 11 sensors-19-02806-f011:**
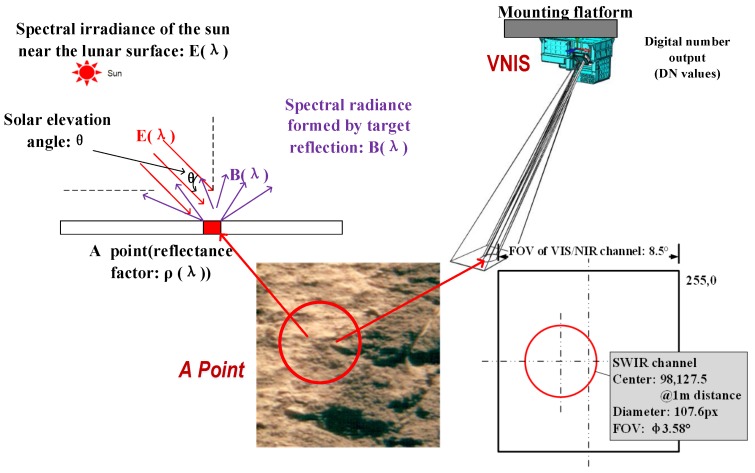
The first scene (which is defined as point A) obtained by VNIS on the far side of the moon. The VNIS is used to detect lunar surface objects and the optical axis of the VIS/NIR channel and SWIR channel are parallel to one another at an 18 mm distance [15,16]. The FOV (Field of view) in the VIS/NIR and SWIR are 8.5° × 8.5° and Φ 3.58°, respectively. The circle represents the SWIR channel’s FOV, which has a diameter of 107.6 pixels and is centered at the coordinate (98, 127.5) of the VIS/NIR image in 1 m detection distance typically.

**Figure 12 sensors-19-02806-f012:**
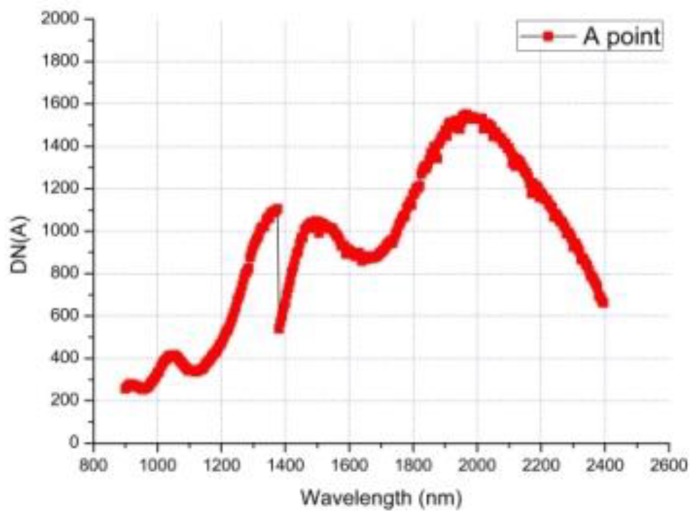
The original digital number (DN) values of the far side of the moon surface at the A point.

**Figure 13 sensors-19-02806-f013:**
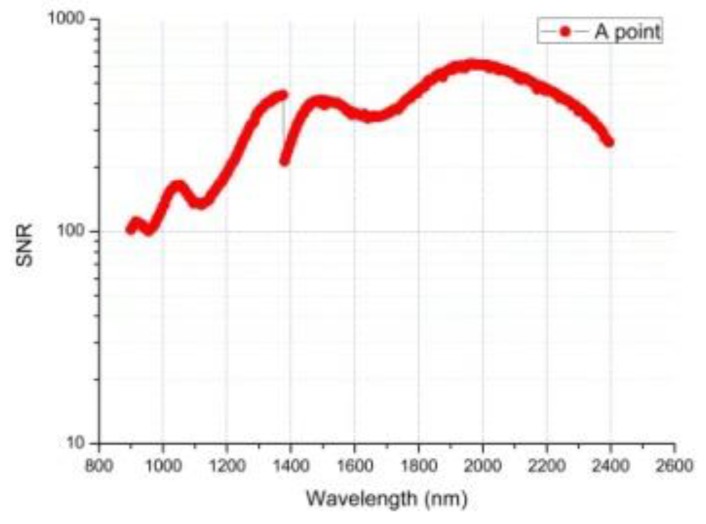
The full-spectrum signal to noise ratio (SNR) curve of the far side of the moon surface at point A.

**Figure 14 sensors-19-02806-f014:**
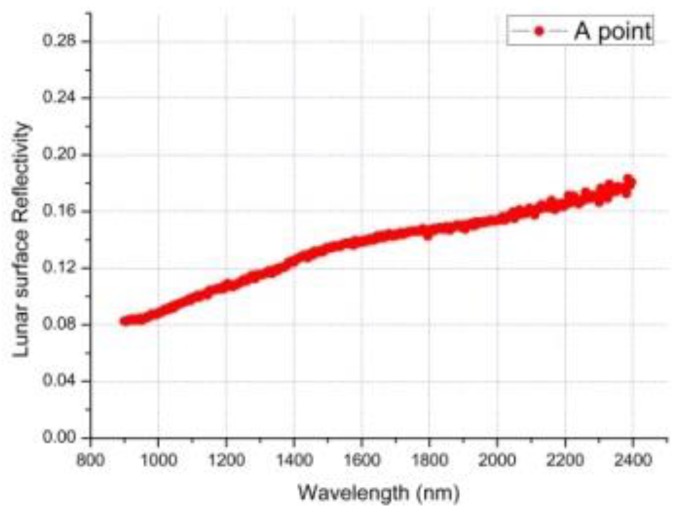
The spectral reflectance curve of the far side of the moon surface at point A.

**Table 1 sensors-19-02806-t001:** The characteristics of the Chang’e-4 visible and near-IR imaging spectrometer (VNIS).

Description	Specification	
	VIS/NIR	SWIR
Spectral range (nm)	450–950	900–2400
Spectral resolution (nm)	2–10	3–12
Number of bands	100	300
Field of view (°)	8.5 × 8.5	ф 3.58
Number of valid pixels	≥256 × 256	1
Quantized value (bit)	10	16
S/N ratio (dB)	≥43 (maximum SNR) ≥33 (albedo 0.09, solar elevation angle 45°)	≥46 (maximum SNR) ≥31 (albedo 0.09, solar elevation angle 15°)
RF ranges (Mhz)	High frequency(F-H): 113.5–177.2 (400–632 nm) Low frequency(F-L): 71.9–113.5 (632–900 nm)	High frequency (F-H): 69.65–117.7 (900–1380 nm) Low frequency (F-L): 42.6–69.65 (1380–2400 nm)
Modulation frequency (Hz)	--	500
Integration time (ms)	18.2–256 (adjustable)	--
Detection range (m)	0.7–1.3	
Measurement time (min)	2	

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
