# Peer review of "The Scientific Information Model of Chang’e-4 Visible and Near-IR Imaging Spectrometer (VNIS) and In-Flight Verification"

_sensors, 2019, doi:10.3390/s19122806_

Reviewer 1 Report

My comments are in the order in which they appear in the manuscript.  Some are editorial suggestions to improve the English language in the writing, while others are related to the paper content.  They are given by line number:

27:  aircraft -> spacecraft

28: an Visible -> a Visible

28: form -> from

36: spectrum -> spectral

43: multicolor -> broadband

49: when an incident light is irradiated to the grating -> when incoming light is incident on the AOTF crystal

67: enters -> enter

71-73: three lights -> three light rays.  This discussion is confusing.  I have worked with AOTFs for several decades and I have never heard the emergent beams referred to as +1 or -1 level diffraction light.  I don't know what the +1 and -1 refers to and this explanation should be reworded.  I suggest reviewing AOTF papers from the 1990s and 2000s (e.g. Georgiev et al. 2002) to get a better understanding of how these diffracted beams should be described.

80: work normal -> change to something more descriptive (and normal is an adjective not an adverb, so it should not be used here in any case).  But what does "work normal" even mean?  It's not clear from this writing.

83-86: this text is largely a repeat of what was written in lines 71-75.  I suggest reorganizing these two sections to improve the flow of the text.

108: remove "will"

109: need "an" or "the" before "AOTF"

118-119: the "great stability" and "high reliability" must be quantified and/or provided references for.

152: the text contains a definition for delta-lambda, but that term does not appear in Equation 1 at all. Is it missing?

174-177: The jump at 1.4 microns in the SWIR channel is not adequately explained. The specifications in Table 1 indicate that the two channels cover 45-950 and 900-2400 nm, so 1400 does not appear to be at the switching point between the two channels.  I cannot tell from this paper whether VNIS uses a single AOTF crystal or whether there are two.  The text implies one, but the figures imply 2.  The paper needs more discussion of the properties of the two AOTFs (manufacturer, physical size, etc).

Author Response

27:  aircraft -> spacecraft  

Response1: Modify as required. The revised text is as follows:

“The Chang’e-4 (CE-4) lunar rover is the first man-made spacecraft launched to the far side of the moon”

28: an Visible -> a Visible

Response2: Modify as required. The revised text is as follows:

“…and its rover (Yutu-2) is equipped with a Visible and Near-IR Imaging Spectrometer(VNIS), which is ...”

28: form -> from

Response3: Modify as required. The revised text is as follows:

“…which is used to analyze the composition of lunar surface minerals, and inherits from the Chang’E-3 Lunar Rover’s Scientific Payloads[1,2].”

36: spectrum -> spectral

Response4: Modify as required. The revised text is as follows:

“…Then the Visible and Near-IR Imaging Spectrometer(VNIS) was powered on and acquired infrared spectral data of the lunar surface..”

43: multicolor -> broadband

Response5: Modify as required. The revised text is as follows:

“…When a beam of broadband light passes through an optically elastic crystal that vibrates at high frequencies.”

49: when an incident light is irradiated to the grating -> when incoming light is incident on the AOTF crystal

Response6: Modify as required. The revised text is as follows:

“…. When incoming light is incident on the AOTF, the wavelength of the diffracted light is related to the frequency of the driving electric signal, and it can be changed by varying the frequency of the electric signal.”

67: enters -> enter

Response7: Modify as required. The revised text is as follows:

“…. The target input rays enter into the instrument through an imaging lens.”

71-73: three lights -> three light rays.  This discussion is confusing.  I have worked with AOTFs for several decades and I have never heard the emergent beams referred to as +1 or -1 level diffraction light.  I don't know what the +1 and -1 refers to and this explanation should be reworded.  I suggest reviewing AOTF papers from the 1990s and 2000s (e.g. Georgiev et al. 2002) to get a better understanding of how these diffracted beams should be described.

Response8: Modify as required. The revised text is as follows:

 “According to the basic principle of AOTF spectrophotometry, after passing through the AOTF, three light rays are formed through the convergent lens, which are positive first-order diffraction light, negative first-order diffraction light and zero-order diffraction light [8]. An InGaAs detector with a diameter of 1mm is placed at the convergence of the positive first-order diffraction light, and a light filter is designed for the zero-order diffraction light, thereby suppressing the stray light.” Figure .4 has also been modified accordingly.

80: work normal -> change to something more descriptive (and normal is an adjective not an adverb, so it should not be used here in any case).  But what does "work normal" even mean?  It's not clear from this writing.

Response9: Modify as required. The revised text is as follows:

 “In order to make the AOTF crystal work in combination with expectations, an radio frequency (RF) signal of a specific frequency(between 42.6MHz to 117.7MHz) should be applied on it, which is generated by the digital display scope (DDS) chip configured with FPGA and then amplified by the RF power amplifier.”

83-86: this text is largely a repeat of what was written in lines 71-75.  I suggest reorganizing these two sections to improve the flow of the text.

Response10: Modify as required. The revised text is as follows:

“After the AOTF crystal is driven by the RF signal, the polychromatic light enters the AOTF penetrates to produce three channels. In the actual design, we choose to detect the positive first-order diffraction light.”

 In addition, the repetitive narrative has been deleted from the article.

108: remove "will"

Response11: Modify as required. The revised text is as follows:

 “As shown in figure. 4, by controlling the output RF signal amplified by the power amplifier, the periodic modulation of the output two channels of monochromatic infrared light at the AOTF outlet can be controlled.”

109: need "an" or "the" before "AOTF"

Response12: Modify as required. The revised text is as follows:

 “As shown in figure. 4, by controlling the output RF signal amplified by the power amplifier, the periodic modulation of the output two channels of monochromatic infrared light at the AOTF outlet can be controlled.”

118-119: the "great stability" and "high reliability" must be quantified and/or provided references for.

Response13: Modify as required. The following elements have been added:

“By controlling the output RF signal amplified by the power amplifier, the model can achieve monochromatic infrared light modulation after AOTF spectrophotometry. The modulation frequency is designed to be 500Hz. This 500Hz frequency is generated by FPGA chip, not by the moving parts(such as motors or galvanometers),it has a great advantage in reliability for space applications.”

152: the text contains a definition for delta-lambda, but that term does not appear in Equation 1 at all. Is it missing?

Response14: Modify as required. Amendments and explanations are as follows:

For the lunar surface spectral detection model, the surface of the moon can be approximated as a Lambert body. The target energy P(Δλ) received by the VNIS can be expressed as[10]:

Where E(λ) represents the spectral irradiance of the sun near the lunar surface. A is the pixel area of detector, F# is the optical aperture, τ0(λ) is the optical system efficiency(include the efficiency of AOTF), θ is the solar elevation angle, Δλ is the spectral resolution, and ρ(λ) is the reflectance factor . Here is also a concept that the actual energy received by the instrument is the spectral radiance B(λ) forced by the target. The relationship between the radiance B(λ) and irradiance E(λ) is[1]:

174-177: The jump at 1.4 microns in the SWIR channel is not adequately explained. The specifications in Table 1 indicate that the two channels cover 45-950 and 900-2400 nm, so 1400 does not appear to be at the switching point between the two channels.  I cannot tell from this paper whether VNIS uses a single AOTF crystal or whether there are two.  The text implies one, but the figures imply 2.  The paper needs more discussion of the properties of the two AOTFs (manufacturer, physical size, etc).

Response15: Modify as required. Amendments and explanations are as follows:

 “The VNIS has two AOTFs, one for VIS/NIR channel, and the other for SWIR channel. Each AOTF is about the size of a matchbox. The two AOTFs were developed by The 26th Research Institute of China Electronic Science and Technology Group Corporation. Each AOTF with two RF connectors, one for high frequency and the other for low frequency. Table 1 also shows the relationship between driving frequencies for AOTFs and the corresponding diffraction wavelength.”

Reviewer 2 Report

Reviewer’s comments on the paper “The scientific information model of Chang’e-4 Visible and Near-IR Imaging Spectrometer (VNIS) and in-orbit verification” by Chunali Li et al.

The paper describes some aspects of the development and verification of the two-channel AOTF spectrometer VNIS, operating at present at the surface (far side) of the Moon onboard Chang’e-4 rover. The Chang’e-4 lunar rover and the VINS instrument represent impressive advancements in space research, and a paper, describing the instrument is highly appreciated and appropriate for Sensors journal. To my knowledge, and judging upon the references list in the paper, this is the first publication about VNIS/Chang’e-4, thus increasing the importance of this publication. 

There are, however, several important caveats, preventing the publication as is. The general comments are:

1)    The paper describes the electronics, signal processing, and SNR of the one instrument’s channel in some detail, mostly ignoring other instrument’s aspects. It is understandable that this is the specific area of work, and the specific contribution, but as it appears to be the first publication on the instrument, a more coherent review of the AOTF, optics, calibration is therefore expected.

2)    The SNR characterisation is incomplete without the calibration description.

3)    The authors cite mostly specific VNIS-related publications (many available only in Chinese), with one exception, even for the basics of the AOTF design and operation. This should be corrected. Some appropriate references may be found in review by Korablev et al. 2018, https://doi.org/10.1364/AO.57.00C103.

4)    English/writing is very difficult in parts. Some terms are not appropriate. Writing noticeably improves starting from page 6, where electronics is described, but still remains perfectible. Though I’ve made some remarks, I am not in position to correct the text throughout, as some paragraphs have to be rewritten completely. I suggest authors to put more work in this, and/or use a professional proofreading service. 

Specific comments:

1)    Heading:  Considerer replacing ‘in-orbit’ with in-flight’, as the instrument sits at the surface.

2)    Line 18, Abstract: signal/noise is redundant and not good. Consider starting the phrase with SNR

3)    Line 27: launched to the far side àbetter to land on the far side

4)    Line 37 : (VINS) was powered on

5)    Paragraph L42-L51 should be modified. AOTF serves not for the light splitting, but for the spectral filtering of light. Splitting is the way to separate the useful filtered signal from the rest of unfiltered light. Please reference basics of the non-collinear AOTF, e.g., Chang 1974. What are ‘spatially periodic modulations’?  To which grating you are referring to?

6)    Table 1 should be appended with all parameters related to the subject of the paper. Some of them are scattered at the following pages. Give here RF ranges, wavelengths subranges, number of spectral bins, modulation frequency/integration time, scan rate, measurement time, etc. 

7)    Introduce the two sub-ranges (two transducers)of the IR channel somewhere here.

8)    L54: remove ‘time dimension’

9)    L56: ~10 µs is not the characteristic time of changing frequency/wavelength, but the characteristic response of the AOTF 

10)L60: Specification should not be capitalized

11)Figure 1: Explain ‘Calibration and dustproof components’ àscanner/positioner/shutter?

12)L72 and further: ‘+1 level’ – usually ‘order’ of diffraction is used. Please mention the polarization of the ±1 orders.

13)L73: ‘When the system is deigned…’ rephrase

14)L87-L88: ‘the two channels…are very weak’ – rephrase

15)Figure 4 carries a little information, while its caption is useful

16)L116: ‘the model can achieve’ – which model?

17)L117: The next sentence is confusing: ‘frequency is designed…, with is a purely electronic method’ – rewrite! Note that the use of AOTF as a modulator is a commonplace.

18)L129: Subject missing

19)L131: ‘one signal is designed’ -- one chain?

20)L112: ‘circuit IS SHOWN in figure 5’

21)L137: ‘8 times of accumulative average…’ is barely comprehensible. Say you stack eight samplings using FPGA

22)L147-148: Remove: ‘transmitted … through the space…’

23)L143: current is generated by light flux, therefore power, not energy. Further L150, spectral irradiance is also in W per cm2, so your P(lambda) is in fact spectral power.

24)Eq. 1: All the variables refer to the optics and are not discussed in the paper. Their values are not given. What is the purpose of giving the equation? Why the solid angle is instantaneous? Delta lambda is defined but not used. Which parameter depends on wavelengths? 

25)Eq. 2: What is A? This is apparently the instrumental response, including the AOTF efficiency, etc. All is hidden in this A, but not explained.

26)L174, Figure 6: The graph relies on something measured, supposedly related to the A variable (Eq.2), but not explained. Here the first time appears the mysterious ‘step’ at 1.4 µm – the transition between two subranges. Please rewrite (see also comment 7).

27)L177: should be ‘600 at 1.7 µm’

28)Para.3.2 Here the lab calibration procedure is to be described, but the description starts from the wrong end. Apparently the filament lamp used is very well chosen, so the signal is approximately equivalent to that expected in flight. Very good, but it proves nothing about the expected SNR. No discussion about the absolute calibration of the laboratory set-up is given, even though Fig 7 shows a laboratory spectrophotometer. The concluding Fig13 is very good, but it could not be obtained without absolute lab or in-flight calibration. No word about that. Give the temperature of the lamp. Discuss the atmospheric absorptions – did you see them in the lab?

29)Fig 9: The waveforms look trivial, but I do not insist.

30)Para 4. Please describe how the absolute reflectance is obtained. Unclear what ref. 8 (Chang’e-3) can tell about the measurements of Chang’e-4. 

31)Conclusions: Do not push the phase-lock principle too much: it is centuries-old, and commonly used with AOTFs. 

Author Response

1)    Heading:  Considerer replacing ‘in-orbit’ with in-flight’, as the instrument sits at the surface.

Response1: Modify as required. The revised text is as follows:

“The scientific information model of Chang’e-4 Visible and Near-IR Imaging Spectrometer(VNIS) and in-flight verification”

“4. In-flight test”…

2)    Line 18, Abstract: signal/noise is redundant and not good. Consider starting the phrase with SNR

Response2: Modify as required. The revised text is as follows:

“The signal characteristic of the VNIS are analyzed in depth, and the signal to noise ratio (SNR) prediction and laboratory verification are presented.”

3)    Line 27: launched to the far side àbetter to land on the far side

Response3: Modify as required. The revised text is as follows:

“The Chang’e-4 (CE-4) lunar rover is the first man-made spacecraft launched to the far side of the moon, and its rover (Yutu-2) is equipped with a Visible and Near-IR Imaging Spectrometer(VNIS), which is used to analyze the composition of lunar surface minerals, and inherits from the Chang’E-3 Lunar Rover’s Scientific Payloads[1,2].”

4)    Line 37 : (VINS) was powered on

Response4: Modify as required. The revised text is as follows:

“Then the Visible and Near-IR Imaging Spectrometer(VNIS) was powered on and acquired infrared spectral data of the lunar surface.”

5)    Paragraph L42-L51 should be modified. AOTF serves not for the light splitting, but for the spectral filtering of light. Splitting is the way to separate the useful filtered signal from the rest of unfiltered light. Please reference basics of the non-collinear AOTF, e.g., Chang 1974. What are ‘spatially periodic modulations’?  To which grating you are referring to?

Response5: Modify as required. The revised text is as follows:

“The VNIS uses an acousto-optic tunable filter (AOTF) for the spectral filtering of light. When a beam of broadband light passes through an optically elastic crystal that vibrates at high frequencies, the monochromatic light of a certain wavelength will be diffracted inside the crystal and transmitted from it at an angle, and the non-diffracted light travels through the crystal directly in the original direction, thereby achieving the goal of light filtering [5].”

6)    Table 1 should be appended with all parameters related to the subject of the paper. Some of them are scattered at the following pages. Give here RF ranges, wavelengths subranges, number of spectral bins, modulation frequency/integration time, scan rate, measurement time, etc.

Response6: Modify as required.

7)    Introduce the two sub-ranges (two transducers)of the IR channel somewhere here.

Response7: Modify as required. Amendments and explanations are as follows:

 “The VNIS has two AOTFs, one for VIS/NIR channel, and the other for SWIR channel. Each AOTF is about the size of a matchbox. The two AOTFs were developed by The 26th Research Institute of China Electronic Science and Technology Group Corporation. Each AOTF with two RF connectors, one for high frequency and the other for low frequency. Table 1 also shows the relationship between driving frequencies for AOTFs and the corresponding diffraction wavelength.”

 This situation is also described in table .1.

8)    L54: remove ‘time dimension’

Response8: Modify as required. The revised text is as follows:

“In this way, the spectral information of the target can be obtained by VNIS [6]. The VNIS can obtain a spectral image in the VIS/NIR channel and spectral data in the SWIR channel simultaneously.”

9)    L56: ~10 µs is not the characteristic time of changing frequency/wavelength, but the characteristic response of the AOTF

Response9: Modify as required. Amendments and explanations are as follows:

“10us” is not the characteristic time of changing frequency or wavelength, but the response time of the AOTF. To avoid misunderstanding , the text deletes this description. The revised text is as follows:

“According to this principle, by changing the driving frequency of AOTF crystal through rapid scanning, the wavelength of the first-order diffraction light passing through the AOTF changes sequentially.”

10)L60: Specification should not be capitalized

Response10: Modify as required.

11)Figure 1: Explain ‘Calibration and dustproof components’ àscanner/positioner/shutter?

Response11:  Modify as required.  Amendments and explanations are as follows:

“The components and basic principle of the VNIS is shown in Figure .1.The VNIS has equipped a calibration unit at the light entrance. The calibration unit consists of ultrasonic motor, framework and diffuser panel inside. The calibration unit could be located at different position by ultrasonic motor for lunar surface detection, in-orbit calibration and dust-proof functions. While the Chang’4 3 spacecraft was operating in-orbit flight and soft landing phase, the calibration unit was locked to thermal insulation and dust-proof. After Yutu-2 rover separated from lander on the moon, VNIS was remote controlled to collect soil images and spectrum and the calibration unit was set fully opened in order not to block the reflectance light into imaging lens. The angle between framework of calibration unit and horizontal mounting face of VNIS was about 55°. When VNIS was required to in-orbit calibration, the calibration unit would be located at horizontal position by Hall sensor and the sun light could be diffused into VNIS from diffuser panel inside. The structure and function of calibration unit is shown in Figure .2. ”

12)L72 and further: ‘+1 level’ – usually ‘order’ of diffraction is used. Please mention the polarization of the ±1 orders.

Response12: Modify as required. Amendments and explanations are as follows:

“According to the basic principle of AOTF spectrophotometry, after passing through the AOTF, three light rays are formed through the convergent lens, which are positive first-order diffraction light, negative first-order diffraction light and zero-order diffraction light [8]. An InGaAs detector with a diameter of 1mm is placed at the convergence of the positive first-order diffraction light, and a light filter is designed for the zero-order diffraction light, thereby suppressing the stray light.”

13)L73: ‘When the system is deigned…’ rephrase

Response13: Modify as required. The revised text is as follows:

“An InGaAs detector with a diameter of 1mm is placed at the convergence of the positive first-order diffraction light, and a light filter is designed for the zero-order diffraction light, thereby suppressing the stray light.”.

14)L87-L88: ‘the two channels…are very weak’ – rephrase

Response14: Modify as required. The revised text is as follows:

“Due to the low albedo of the lunar surface, the positive first-order diffraction light after AOTF is very weak.”

15)Figure 4 carries a little information, while its caption is useful

Response15: Modify as required.

16)L116: ‘the model can achieve’ – which model?

Response16: Modify as required. The revised text is as follows:

 “By controlling the output RF signal amplified by the power amplifier, the periodic modulation of the output positive first-order diffraction light at the AOTF outlet can be controlled. The modulation frequency is designed to be 500Hz.”

17)L117: The next sentence is confusing: ‘frequency is designed…, with is a purely electronic method’ – rewrite! Note that the use of AOTF as a modulator is a commonplace.

Response17: Modify as required. The revised text is as follows:

“By controlling the output RF signal amplified by the power amplifier, the model can achieve monochromatic infrared light modulation after AOTF spectrophotometry. The modulation frequency is designed to be 500Hz.”

18)L129: Subject missing

Response18: Modify as required.

19)L131: ‘one signal is designed’ -- one chain?

Response19: Modify as required.

“Specifically, one chain is designed as a voltage follower circuit and the other is designed as a reverse circuit, thus realizing a phase shift of 180 degrees.”

20)L112: ‘circuit IS SHOWN in figure 5’

Response20: Modify as required. The revised text is as follows:

“Figure. 5 shows the picture of the SWIR channel processing circuit and the InGaAs infrared detector”

21)L137: ‘8 times of accumulative average…’ is barely comprehensible. Say you stack eight samplings using FPGA

Response21: Modify as required. The revised text is as follows:

“In order to further improve the detection sensitivity, the signal of the same wavelength is collected 8 times by ADC chip(AD976), and then the sampled DN values of 8 times are averaged are conducted within the FPGA.”.

22)L147-148: Remove: ‘transmitted … through the space…’

Response22: Modify as required.

23)L143: current is generated by light flux, therefore power, not energy. Further L150, spectral irradiance is also in W per cm2, so your P(lambda) is in fact spectral power.

Response23: Modify as required. The revised text is as follows:

For the lunar surface spectral detection model, the surface of the moon can be approximated as a Lambert body. The target energy P(Δλ) received by the VNIS can be expressed as[10]:

Where E(λ) represents the spectral irradiance of the sun near the lunar surface. A is the pixel area of detector, F# is the optical aperture, τ0(λ) is the optical system efficiency(include the efficiency of AOTF), θ is the solar elevation angle, Δλ is the spectral resolution, and ρ(λ) is the reflectance factor . Here is also a concept that the actual energy received by the instrument is the spectral radiance B(λ) forced by the target. The relationship between the radiance B(λ) and irradiance E(λ) is[1]:

24)Eq. 1: All the variables refer to the optics and are not discussed in the paper. Their values are not given. What is the purpose of giving the equation? Why the solid angle is instantaneous? Delta lambda is defined but not used. Which parameter depends on wavelengths?

Response24: Modify as required.

25)Eq. 2: What is A? This is apparently the instrumental response, including the AOTF efficiency, etc. All is hidden in this A, but not explained.

Response25: Modify as required. The revised text is as follows:

Generally, the target energy P(λ) is determined after the instrument system parameters are determined. That is, given the target radiant power received by the InGaAs detector in the information flow model in figure. 6, the signal current I(λ) of the infrared detector response can be expressed as:

Where Rλ represents the current response of detector. The VNIS uses the J23TE2-66C-R01M-2.6 infrared detector manufactured by Judson, and the peak current response rate is about 1.2A/W.

….

The parameters used in the calculations are:

Ø  E=217W/m2/μ[email protected]μm;

Ø  A=1mm2;

Ø  F#:2.8;

Ø  τ0 [email protected]μm;

Ø  Δλ[email protected]μm;

Ø  Rλ=1.2A/[email protected]μm;

Ø  θ=15°;

Ø  ρ[email protected]μm;

Ø  B=1.61W/m2/μm/[email protected]μm.

26)L174, Figure 6: The graph relies on something measured, supposedly related to the A variable (Eq.2), but not explained. Here the first time appears the mysterious ‘step’ at 1.4 µm – the transition between two subranges. Please rewrite (see also comment 7).

Response26: Modify as required. The revised text is as follows.:

 “As shown in figure. 7, there exists a step at 1380nm. The reason is that the SWIR AOTF crystal has two driving frequency range(F-H and F-L), and the 1380nm is exactly the frequency switching point of SWIR AOTF crystal.”

27)L177: should be ‘600 at 1.7 µm’

Response27: Modify as required. The revised text is as follows.:

“It can be seen from the figure 7 that the SNR is about 600 at 1.7μm”

  28)Para.3.2 Here the lab calibration procedure is to be described, but the description starts from the wrong end. Apparently the filament lamp used is very well chosen, so the signal is approximately equivalent to that expected in flight. Very good, but it proves nothing about the expected SNR. No discussion about the absolute calibration of the laboratory set-up is given, even though Fig 7 shows a laboratory spectrophotometer. The concluding Fig13 is very good, but it could not be obtained without absolute lab or in-flight calibration. No word about that. Give the temperature of the lamp. Discuss the atmospheric absorptions – did you see them in the lab?

Response28: Modify as required. The revised text is as follows:

After the information flow model is established, the signal characteristic and the SNR of VNIS are tested in the laboratory. Figure. 8 is the photo of laboratory testing, where the light source is a halogen lamp and the target is a diffuse reflection plate. Adjusting the intensity of the Halogen lamp can simulate the change of the spectral radiance forced by the diffuse reflection plate, and synchronous measurement is carried out by a commercial spectrometer(FieldSpec 4).

When the spectral radiance at 1.7μm measured by the FieldSpec 4 is about 1.61W/m2/μm/Sr, which is equal to the value discussed in Section 3.1, the intensity of the halogen lamp is stabilized and the test is started. Figure. 9 shows the original digital number (DN)values of 300 infrared spectral bands, and the SNR is about [email protected]μm (the signal DN is 1600, and the noise DN is about 3.2). This is slightly different from the value (SNR≈[email protected]μm) given in Section 3.1, because the light source here is not the sunlight with an equivalent elevation angle of 15 degrees and some transmission errors caused by the commercial spectrometer.

29)Fig 9: The waveforms look trivial, but I do not insist.

Response29: Modify as required.

30)Para 4. Please describe how the absolute reflectance is obtained. Unclear what ref. 8 (Chang’e-3) can tell about the measurements of Chang’e-4.

Response30: Modify as required. Amendments and explanations are as follows:

“Through absolute lab and in-flight radiometric calibration, we can get the relationship between DN values and spectral radiance at the entrance pupil, which is consistent with B(λ) in Figure.10. The spectral radiance at the entrance pupil B(λ) divided by the solar irradiance E(λ), multiplied by π, and further divided by the cosine of the incident angle θ can obtain the reflectance factor ρ(λ) [1].”

31)Conclusions: Do not push the phase-lock principle too much: it is centuries-old, and commonly used with AOTFs.

Response31: Modify as required. The revised text is as follows:

“In view of the extremely weak infrared reflectance spectral signals on the lunar surface, an information flow model, on the basis of which the signal-to-noise ratio has been predicted. As of the end of February 2019, the VNIS has operated 6 times on the lunar surface, and acquired 12 sets of infrared spectral data. It is found that an average SNR of 300 can still be obtained when the lunar albedo is around 9% and the solar elevation angle is 15 degrees, which further verifies the effectiveness of the proposed method for weak infrared signal. The SNR values of in-flight testing are basically consistent with the predictions given in this paper. In China’s future deep space exploration programs, the method proposed in this paper will be helpful to further study the acquisition of weak infrared spectral information on the surface of planets.”

Round  2

Reviewer 2 Report

The authors have corrected the manuscript and they have adequately addressed most of the comments. The only remaining comment is the quality of the introduction, and referencing of other's (not related to the Chinese spacecraft AOTF spectrometers) work. This was not done. The first reviewer has also suggested to consult Georgiev et al. 2002. I fully concur and suggest that type of work should be consulted and cited. 

Once this is done, I feel the paper can be published, and I do not need to see it another time.

Author Response

Chapter I has been amended to read as follows:

1. Introduction

The Chang’e-4 (CE-4) lunar rover is the first man-made spacecraft launched to the far side of the moon, and its rover (Yutu-2) is equipped with a Visible and Near-IR Imaging Spectrometer(VNIS), which is used to analyze the composition of lunar surface minerals [1,2]. The VNIS inherits from the Chang’E-3 Lunar Rover’s Scientific Payloads, and the software is optimized. The VNIS is a spectrum detector based on AOTF spectroscopy[3,4], and it has two detection channels - visible near-infrared(450~950nm) and short-wave infrared(900~2400nm). It performs spectral analysis and imaging detection of minerals on the lunar surface under appropriate solar illumination, and assists in the comprehensive detection of minerals and chemical compositions in the patrol areas.

AOTF spectrometer has been used in deep space exploration for a long time. The SPICAM[5], which is Loaded on the ESA Mars Express mission(2003), is one of the first spectroscopic applications of the AOTF on a civilian spacecraft. It’s a single-pixel spectrometer which can sequentially measure the spectrum of reflected solar radiation from Mars between 1.1 and 1.7 µm with spectral resolution of 3.5 cm-1, and spatial resolution of ~6 km from 250km Mars Express orbit.  Similar instruments SPICAV[6] were loaded on the ESA Venus Express mission(2005). NASA has also been working on spectrometers based on the AOTF. AIMS[7] is a compact, two-channel AOTF imaging spectrometer developed by NASA/GSFC, and it is one of the alternative payloads of NASA Mars rover. Since 2006, Shanghai Institute of Technical Physics began to study the imaging spectrometer based on AOTF. VNIS is a payload of lunar rover for 0.45-2.4μm spectral bands detection and will be able to inspect and probe mineral for rover region[2] (China’s Chang’e 3 and Chang’e 4 unmanned lunar rover).

The CE-4 spacecraft was successfully launched on December 8, 2018, and it landed on the moon on January 3, 2019. Then the VNIS was powered on and acquired infrared spectral data of the lunar surface. This paper focuses mainly on simulation analyses and laboratory tests based on the infrared channel design features and information flow model of the instrument, and carries out verification according to the acquired in-orbit data.
